

# Updated results on the cosmic ray energy spectrum and composition from the GRAPES-3 experiment

F. Varsi[1⋆], S. Ahmad[2], M. Chakraborty[3], A. Chandra[2], S. R. Dugad[3], U. D. Goswami[4], S. K. Gupta[3], B. Hariharan[3], Y. Hayashi[5], P. Jagadeesan[3], A. Jain[3], P. Jain[1], S. Kawakami[5], H. Kojima[6], S. Mahapatra[7], P. K. Mohanty[3], R. Moharana[8], Y. Muraki[9], P. K. Nayak[3], T. Nonaka[10], A. Oshima[6], B. P. Pant[8], D. Pattanaik[3,7], G. Pradhan[11], M. Rameez[3], K. Ramesh[3], L. V. Reddy[3], R. Sahoo[11], R. Scaria[11], S. Shibata[6], K. Tanaka[12] and M. Zuberi[3]

**1** Indian Institute of Technology Kanpur, Kanpur 208016, India
**2** Aligarh Muslim University, Aligarh 202002, India
**3** Tata Institute of Fundamental Research, Homi Bhabha Road, Mumbai 400005, India
**4** Dibrugarh University, Dibrugarh 786004, India
**5** Graduate School of Science, Osaka City University, Osaka 558-8585, Japan
**6** College of Engineering, Chubu University, Kasugai, Aichi 487-8501, Japan
**7** Utkal University, Bhubaneswar 751004, India
**8** Indian Institute of Technology Jodhpur, Jodhpur 342037, India
**9** Institute for Space-Earth Environmental Research, Nagoya University, Nagoya 464-8601, Japan
**10** Institute for Cosmic Ray Research, Tokyo University, Kashiwa, Chiba 277-8582, Japan
**11** Indian Institute of Technology Indore, Indore 453552, India
**12** Graduate School of Information Sciences, Hiroshima City University, Hiroshima 731-3194, Japan

⋆ fahim@iitk.ac.in

## Abstract

Here, we present the updated results on the cosmic ray energy spectrum and composition analysis from the GRAPES-3 experiment over the energy range of 50 TeV to 1000 TeV since ICRC 2021. The simulation of cosmic ray showers was performed using the post-LHC high energy hadronic interaction model QGSJetII-04 and low energy hadronic model FLUKA. A detailed GEANT4 simulation of the GRAPES-3 muon telescope was performed. The composition was obtained by fitting simulated muon multiplicity distributions for proton, helium, nitrogen, aluminium, and iron primaries to the observed data. The energy spectrum connects to the direct measurements with a fairly good agreement in flux.

# 1 The GRAPES-3 experiment

The GRAPES-3 (acronym: Gamma Ray Astronomy at PeV Energies Phase-3) experiment is located at Ooty, South India (11.4°N latitude, 76.7°E longitude, and 2200 m altitude above *m.s.l.*). It consists of a most compact array of 400 plastic scintillator detectors, each with an area of 1 m² and an inter-detector separation of 8 m, spreading over a total area of 25000 m² [1–3]. Being a highly dense EAS array, the GRAPES-3 experiment can observe the primary cosmic rays (PCRs) in TeV-PeV energy range, providing a significant overlap with direct experiments [4–6]. It also contains 16 similar independent muon modules, forming the largest muon telescope (G3MT) with an area of 560 m² [7]. The GRAPES-3 muon telescope (G3MT) has an energy threshold of $\sec\theta$ GeV, where $\theta$ is the zenith angle of the incident muon. The muon multiplicity distribution observed by the G3MT is sensitive to PCRs mass and an important parameter for the precise measurements of the PRCs composition [8].

# 2 MC simulations

The CORSIKA package version 7.6900 was used to simulate the extensive air showers (EAS) for proton (H), helium (He), nitrogen (N), aluminium (Al) and iron (Fe), using the QGSJET-II-04 and FLUKA models as high and low energy hadronic interaction models, respectively. For each simulated primary, nearly $6\times10^7$ EAS were simulated in an energy range of 1 TeV - 10 PeV divided in 20 equal logarithmic bins of a width of 0.2. The number of EAS simulated in each energy bin is listed in Table 1. In each energy bin, EAS were generated in zenith angle range of 0° - 45°, assuming a power law of differential spectral index −2.5. For the analysis, each EAS was thrown ten times in a circular area of radius 150 m from the center (−13.85 m, 6.29 m) of the GRAPES-3 scintillator detectors array with a random core position to improve the statistics. The CORSIKA simulation output was further analyzed to obtain particle density, using a detailed simulation of the GEANT4 response of $\gamma$-rays, electrons, muons, and hadrons in the plastic scintillator detectors. A detailed GEANT4 simulation of the response of the secondaries in the G3MT had also been performed. Since the top concrete of G3MT absorbs the electromagnetic component of the EAS, the GEANT4 response of muons and hadrons was measured in terms of PRC hits, using a discriminator threshold of 0.2 times of the energy deposited by the minimum ionizing particle (∼4 keV). This PRCs hit status was used to reconstruct the muon tracks (statistical measure of the muons).

# 3 Reconstruction procedure, quality cuts and data summary

For each triggered EAS, the parameters such as core location $(X_c, Y_c)$, size $(N_e)$, and age $(s)$ are obtained by minimizing the observed lateral distribution of particles densities in different scintillator detectors with a lateral density distribution function, namely Nishimura-Kamata-Greisen (NKG) as given below, based on a log-likelihood method using TMINUIT package,

$$\rho(r_i) = \frac{N_e}{2\pi r_M^2} \frac{\Gamma(4.5-s)}{\Gamma(s)\Gamma(4.5-2s)} \left(\frac{r_i}{r_M}\right)^{(s-2)} \left(1+\frac{r_i}{r_M}\right)^{(s-4.5)}, \qquad (1)$$

where $\rho(r_i)$ is the particle density observed by $i^{th}$ detector, $r_i$ is the lateral distance of $i^{th}$ detector from shower core, and $r_M$ is the Moliere radius which is 103 m for the GRAPES-3 observational site. The initial estimate of arrival direction $(\theta, \phi)$ is measured by fitting the relative arrival time of secondary particles recorded by different scintillator detectors with a

Table 1: Summary of the number of EAS simulated using CORSIKA for each of the PCRs (H, He, N, Al, and Fe).

| Bin | Energy range [TeV] | Number of EAS | Bin | Energy range [TeV] | Number of EAS |
|-----|-------------------|---------------|-----|-------------------|---------------|
| 1 | 1.00 - 1.58 | $2.5 \times 10^7$ | 11 | 100.00 - 158.49 | $1.0 \times 10^5$ |
| 2 | 1.58 - 2.51 | $1.5 \times 10^7$ | 12 | 158.49 - 251.19 | $5.0 \times 10^4$ |
| 3 | 2.51 - 3.98 | $1.0 \times 10^7$ | 13 | 251.19 - 398.11 | $2.5 \times 10^4$ |
| 4 | 3.98 - 6.31 | $5.0 \times 10^6$ | 14 | 398.11 - 630.96 | $1.5 \times 10^4$ |
| 5 | 6.31 - 10.00 | $2.5 \times 10^6$ | 15 | 630.96 - 1000.00 | $1.0 \times 10^4$ |
| 6 | 10.00 - 15.85 | $1.5 \times 10^6$ | 16 | 1000.00 - 1584.89 | $5.0 \times 10^3$ |
| 7 | 15.85 - 25.12 | $1.0 \times 10^6$ | 17 | 1584.89 - 2511.89 | $2.5 \times 10^3$ |
| 8 | 25.12 - 39.81 | $5.0 \times 10^5$ | 18 | 2511.89 - 3981.07 | $1.5 \times 10^3$ |
| 9 | 39.81 - 63.10 | $2.5 \times 10^5$ | 19 | 3981.07 - 6309.57 | $1.0 \times 10^3$ |
| 10 | 63.10 -100.00 | $1.5 \times 10^5$ | 20 | 6309.57 - 10000.00 | $5.0 \times 10^2$ |

plane front, followed by correction for the shower front curvature based on shower size and age to obtain a more accurate EAS direction [9].

To reduce the systematics cause by EAS reconstruction, event selection criteria are applied. The EAS having successful parameters and arrival direction reconstruction are used in the analysis. Shower cores are restricted to a circular area of 50 m from the center of the array to minimise the contamination of the improper reconstruction due to EAS with core landing near the edge or outside the array. The shower age is restricted between 0.2 and 1.8 to avoid improper reconstruction due to shower age converging to its limits. The zenith angle is retricted to 18° to minimize the systematics due to inclined EAS. To ensure the trigger efficiency > 90%, the EAS having shower size $\geq 10^4$ are used.

Data collected during 1 January 2014 - 26 October 2015 ($\sim$ 22 months) is used for the analysis. The total live time of data collection is $\sim$463 days. The number of showers remaining after applying all the quality cuts are $1.47 \times 10^7$ from a total set of $1.75 \times 10^9$ EAS.

## 4 Analysis

For each simulated primary, the trigger ($\varepsilon_T$) and reconstruction ($\varepsilon_R$) efficiencies are calculated as a function of primary energy, and total efficiency ($\varepsilon_{tot}$) is determined by the product of trigger and reconstruction efficiencies. Acceptance($A_{acc}$) is represented as the product of the effective area and the effective viewing angle as,

$$A_{acc}(E_T) = \frac{\pi A}{2} \sum_{k=1}^{n_\theta} \varepsilon_{tot}(E_T, \theta_k)(\cos 2\theta_k - \cos 2\theta_{k+1}),  \qquad (2)$$

where $A$ is the fiducial area, $n_\theta$ is the total number of angle bins and $\theta_k$ and $\theta_{k+1}$ are low and high edges of each angle bin, respectively. The trigger efficiency and acceptance of all simulated primaries are shown in Figure 1. The trigger efficiency is >90% at 40 TeV, 45 TeV, 60 TeV, 70 TeV and 85 TeV for H, He, N, Al and Fe, respectively. The acceptance is increased to 2300 m$^2$ sr at 100% efficiency for $\theta < 18°$.

The G3MT is dedicated to measure the number of muon tracks for each triggered shower. With an increase in the number of incident muons in a given module, the muon tracks start to overlap, resulting in the underestimation of reconstructed tracks. This saturation effect, for one module, is shown in the left panel of Figure 2 for simulations. The curve is modelled with

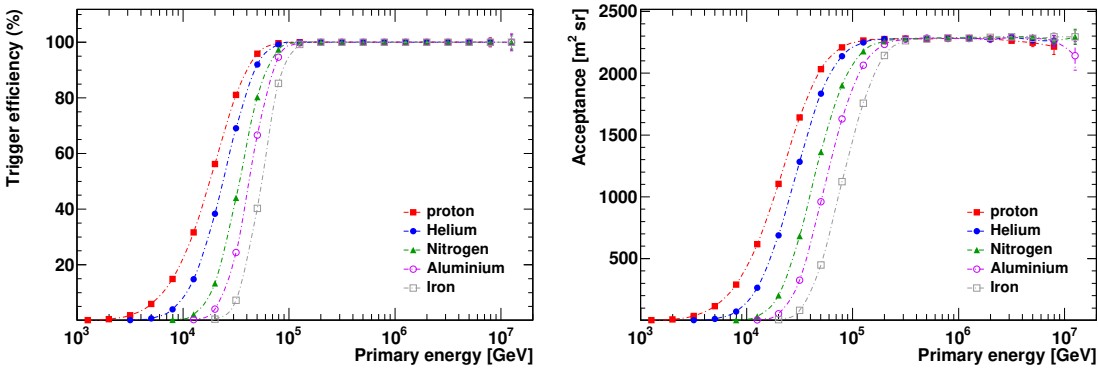

Figure 1: Trigger efficiency (left) and acceptance of GRAPES-3 scintillator detectors array (right) as a function of primary energy for $\theta < 18°$.

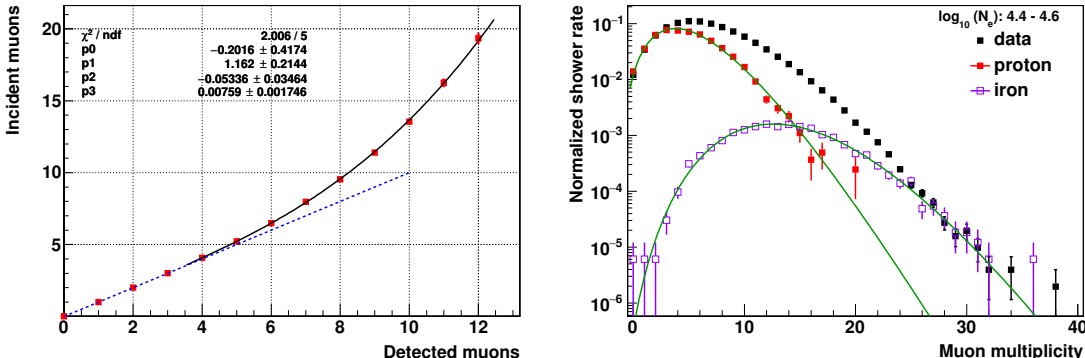

Figure 2: Left: Muon saturation for one module of the G3MT, modelled with third-order polynomial represented by the black curve. The blue dashed line guides the linearity between the incident and detected muons. Right: Muon multiplicity of H and Fe fitted with the negative binomial distribution, plotted along with observed MMD for $4.4 \leq \log(N_e) < 4.6$.

a third-order polynomial for detected muons $\geq 4$, which is used to get the correct estimate of the muon multiplicity for both simulation and observed data. The shape of muon multiplicity distribution is well described by the negative binomial distribution (NBD) and is given by,

$$NBD\left(x; m, \sigma^2\right) = \frac{\Gamma\left(x + \frac{m^2}{\sigma^2 - m}\right)}{\Gamma(x+1)\Gamma\left(\frac{m^2}{\sigma^2 - m}\right)} \left(\frac{m}{\sigma^2}\right)^{\frac{m^2}{\sigma^2 - m}} \left(\frac{\sigma^2 - m}{\sigma^2}\right)^x, \tag{3}$$

where $x$ is the muon multiplicity value and $m$ and $\sigma^2$ are the mean value and standard deviation of muon multiplicity distributions (MMDs), repectively. Thus, the normalized MMD of each simulated primary is fitted with NBD to model statistical fluctuations and shown in the right panel of Figure 2 for proton and iron along with observed MMD, for $4.4 \leq \log(N_e) < 4.6$. The distributions of proton and iron are scaled such that the tails of distributions overlap with the observed MMD. The low and high multiplicity of observed MMD is well described by the proton and iron, respectively. However, primaries of intermediate mass group are required to describe the middle range of the observed MMD. The unfolding method is used to extract the relative composition of the simulated primaries group for each shower size bin, using the Gold's algorithm. The observed muon multiplicity vector ($\mu$) contains the observed MMDs and the response matrix $R$ contains the probability values such that $R_{i,j}$ represents the probability

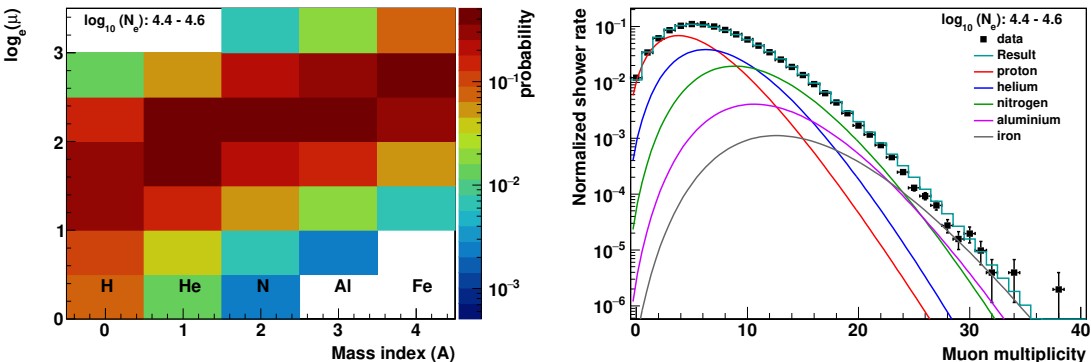

Figure 3: Left: Response matrix **R** for the MMD for the EAS induced by all simulated primaries. Right: Comparision of the resultant MMD with the observed MMD after scaling the simulated MMD with the relative compositional weights; for $4.4 \leq \log(N_e) < 4.6$.

of an EAS initiated by the $j^{th}$ simulated primary having the muon multiplicity value $\mu_i$. The response matrix for $4.4 \leq \log(N_e) < 4.6$ is shown in the left panel of Figure 3, where the color gradient represents the probability values. The Gold algorithm is an iterative method and the relative composition vector for $(k+1)^{th}$ iteration $(A^{k+1})$ is estimated from the estimate of the $k^{th}$ iteration $(A^k)$ as,

$$A^{k+1} = \frac{A^k (CR)^T (C\mu)}{(CR)^T (CR) A^k},$$

(4)

where **C** is the error matrix for the observed data such that $C_{i,j} = \delta_{i,j}/\sqrt{\mu_i}$. The relative composition assumed by the H4a models for the all simulated primaries used as the prior for the unfolding. The algorithm tries to minimize the $\chi_k^2$ for the muon multiplicity vector $((CR)^T C\mu)$ and the forward unfolded muon multiplicity vector $((CR)^T (CR) A^k)$. The number of iterations are stoped where the chi-square improvement ($\Delta \chi_k^2 = \chi_k^2 - \chi_{k-1}^2$) becomes less than $10^{-3}$. The result of the unfolding is shown in the right panel of the Figure 3 for $4.4 \leq \log(N_e) < 4.6$, where each NBD curve corresonding to the simulated primaries is scaled with its relative composition and the resulting MMD exhibiting a good agreement with the observed MMD.

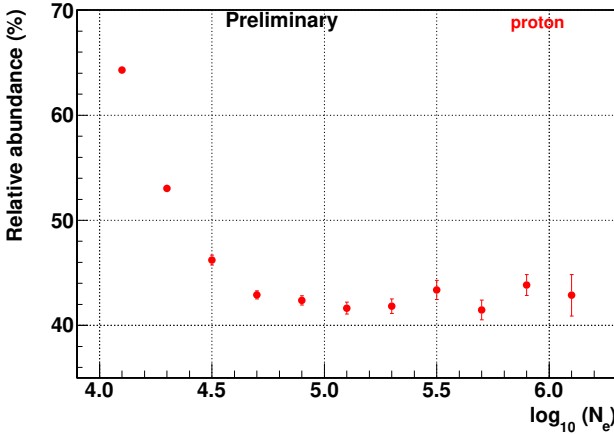

Figure 4: Relative composition of proton as a function of shower size.

# 5  Results

The preliminary results for the measurements of relative composition and energy spectrum for proton component are presented. The relative abundance of the proton component decreases from ∼65% at shower size $10^{4.1}$ to ∼42% at shower size $10^{6.1}$ and is shown in the left panel of Figure 4. In order to get the size distribution for the proton component, the fraction of the number of showers is selected from the observed size distribution based on the relative abundance of the proton component for each shower size bin in the shower size range from $10^{4.0}$ to $10^{6.2}$. Gold's algorithm is used to unfold the proton energy distribution from corresponding size distribution, assuming the H4a proton spectrum as the prior. The differential cosmic-ray spectrum (dI/dE) can be expressed as follows,

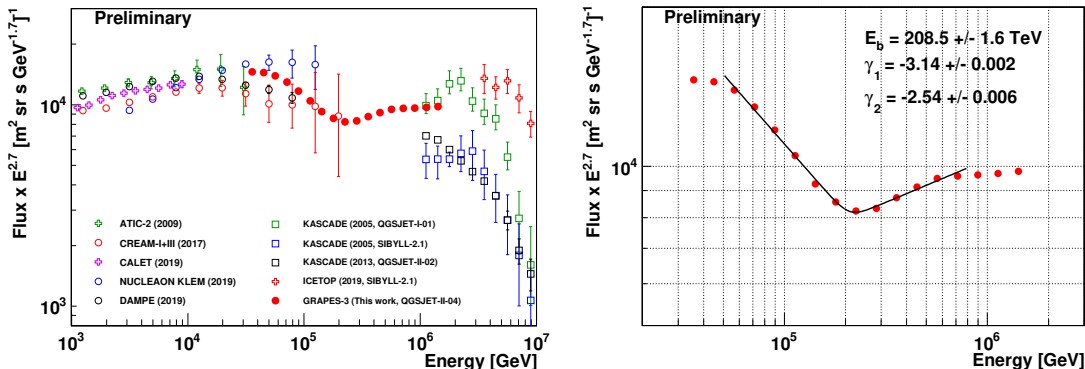

Figure 5: Left: Elemental spectrum of proton; compared with other experiments [4–6, 10–14]. Right: Elemental spectrum of proton fitted with a smoothly broken power law (solid curve) showing an energy break at ∼210 TeV.

$$\frac{dI}{dE} = \frac{1}{T_{obs}}\left(\frac{N}{\Delta E \cdot A_{acc}}\right)_i , \tag{5}$$

where $N$, $A_{acc}$ and $\Delta E$ are the number of EAS, acceptance and width of $i^{th}$ energy bin, respectively, and $T_{obs}$ is the live time of the data. The measured preliminary energy spectrum of proton plotted with direct [4–6, 10, 11] and indirect observations [12–14] is shown in the left panel of Figure 5. The statistical error bars are smaller than the marker size. The flux of the measured proton spectrum in this work is consistent with CREAM I+III at lower energy and is consistent with KASCADE (QGSJET-I-01) at higher energy. The measured proton spectrum is fitted with a broken power law in an energy range from 50 TeV to 800 TeV as shown in the right panel of Figure 5. An energy break ($E_b$) is observed at nearly 210 TeV with the spectral slope of −3.14 and −2.54 before ($\gamma_1$) and after ($\gamma_2$) the energy break.

# Acknowledgements

We are grateful to D.B. Arjunan, A.S. Bosco, V. Jeyakumar, S. Kingston, N.K. Lokre, K. Manjunath, S. Murugapandian, S. Pandurangan, B. Rajesh, R. Ravi, V. Santoshkumar, S. Sathyaraj, M.S. Shareef, C. Shobana, R. Sureshkumar for their role in efficient running of the experiment

**Funding information**  We acknowledge support of the Department of Atomic Energy, Government of India, under Project Identification No. RTI4002. This work was partially supported by grants from Chubu University, ISEE of Nagoya University and ICRR of Tokyo University,

Japan. Numerical computations were in part carried out on PC cluster at Center for Computational Astrophysics, NAOJ, Japan. The first author is thankful to the University Grant Commission, New Delhi, for the support of his fellowship.

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
