# Peer review of "Updated results on the cosmic ray energy spectrum and composition from the GRAPES-3 experiment"

_SciPost Physics Proceedings, doi:SciPost Phys. Proc. 13, 031 (2023)_

## Round 1 · Referee Report · Anonymous (Referee 1) · 2022-9-6

Strengths

- The article is well written and mostly clear. The analysis is sophisticated for an intermediate result.

Weaknesses

- The paper itself is good, but the analysis can be improved. This has nothing to with the quality of the paper.
- The analysis splits the problem of unfolding the spectra of five cosmic ray components into several consecutive steps, but this is only an approximation. The state-of-the-art is to perform a 2D unfolding that takes muon multiplicity and Ne into account, similar to what KASCADE did. Doing so may change the results.

Report

The journals criteria are met. I found a few points that require clarification, but nothing major. I have a few comments on the methods to the authors which are merely informative, and a few optional suggestions. I am sure that the authors are aware of most points that I raise and I understand that the current state of the analysis is not the final one.

Requested changes

- page 2: "spreading over a total area of 25000 m2" Suggestion: It may be useful to give a size comparison to a comparable array, perhaps KASCADE
- page 2: "providing a significant overlap with direct experiments" Suggestion: List important other direct experiments which have overlap with GRAPES-3.
- page 2: "using a discriminator threshold of 0.2 particles (∼4 keV)" Please clarify whether 0.2 particles refers to vertical equivalent muon (VEM) or minimum ionising particle deposit (MIPS) or something else.
- page 2: "namely Nishimura-Kamata-Greisen (NKG) as given below" Comment: as far as I remember, the NKG formula was derived for electrons, not muons, and thus does not fit as well to the muon profile than other more empirical formulas.
- page 3: "To ensure the trigger efficiency > 90%, the EAS having shower size ≥ 104 are used." Comment: Using a cut on a reconstructed variable like the fitted shower size introduces a "survival bias". This distorts the measurement. The general principle is discussed in https://inspirehep.net/literature/1406466 (see Fig. 4) or https://inspirehep.net/literature/1357170 (see Fig. 1), although the detailed situation in the second reference is not directly comparable to GRAPES.
- page 4: "the mean value and standard deviation of MMD" the acronym MMD is not introduced
- page 4: "the probability vaules such that" values
- page 4: "The Gold algorithm is an iterartive method" iterative
- page 4: Comment regarding unfolding: the response kernel R should model both the efficiency of the detector and its resolution effects to be consistent. It is less likely that results are correct when the efficiency is handled in a separate analysis step.
- Fig 2: Comment: I suggest to swap x and y axis and fit the correction curve detected muons(incident muons). This seems like more work at first glance, because you then need to invert this function to obtain incident muons(detected muons). However, even more correct is to include this effect into the response kernel R during unfolding instead of doing a point-to-point correction. The presented approach seems like it is easier to get the correction function directly, but this introduces an issue. "Incident muons" is the truth and the measured "Detected muons" has uncertainties. A typical least-squares fit cannot deal with uncertainties along the x-axis where the measurement is currently placed. In summary, this is not only a cosmetic change.
- Fig 3: Comment: I understand this analysis is not final, but I suggest to eventually follow KASCADE and fit the 2D distributions in muon multiplicity and Ne. The reason is that Ne is not a very precise energy estimator and not independent of the cosmic ray mass. A more correct approach is to do an unfolding in 2D in which muon multiplicity and Ne are used as input to get the energy spectra of the five components as output.
- page 5: "The number of iterations are stoped where the rate of convergence ... is less than" stopped when the chi-square improvement ... becomes less then
- page 5: "with its relative composition and the resultant MMD" Perhaps "resulting"?
- page 5: "The relative abundance of the proton" of protons (or "of the proton component"); also change in the following sentences likewise
- page 5: "The Gold’s algorithm" Remove "The"
- page 5: "The relative abundance of proton is used to decouple the corresponding size distribution from the observed size distribution in the shower size range from 104.0 to 106.2." It is not clear what exactly is done here, please elaborate. As mentioned previously, it is an approximation to unfold the muon multiplicity distribution and the energy distribution separately, since this is neglects correlations between the two.
- Fig 5: Comment: The break in the proton spectrum is a very interesting result, that potentially has consequences for people who perform fits of the cosmic ray flux and its composition (e.g. GSF).

---

## Editorial Decision

published